# CoRec: An Easy Approach for Coordination Recognition

**Qing Wang, Haojie Jia, Wenfei Song, Qi Li**
Department of Computer Science, Iowa State University, Ames, Iowa, USA
{qingwang, hjjia, wsong, qli}@iastate.edu

## Abstract

In this paper, we observe and address the challenges of the coordination recognition task. Most existing methods rely on syntactic parsers to identify the coordinators in a sentence and detect the coordination boundaries. However, state-of-the-art syntactic parsers are slow and suffer from errors, especially for long and complicated sentences. To better solve the problems, we propose a pipeline model COordination RECognizer (CoRec). It consists of two components: coordinator identifier and conjunct boundary detector. The experimental results on datasets from various domains demonstrate the effectiveness and efficiency of the proposed method. Further experiments show that CoRec positively impacts downstream tasks, improving the yield of state-of-the-art Open IE models. Source code is available[1].

## 1 Introduction

Coordination is a widely observed syntactic phenomenon in sentences across diverse corpora. Based on our counting, 39.4% of the sentences in OntoNotes Release 5.0 (Weischedel et al., 2013) contain at least one coordination. The frequently appeared conjunctive sentences bring many challenges to various NLP tasks, including Natural Language Inference (NLI) (Saha et al., 2020), Named Entity Recognition (NER) (Dai et al., 2020), and text simplification (Xu et al., 2015). Specifically, in Open Information Extraction (Open IE) tasks, researchers find that ineffective processing of conjunctive sentences will result in substantial yield lost (Corro and Gemulla, 2013; Saha and Mausam, 2018; Kolluru et al., 2020), where yield is essential since Open IE tasks aim to obtain a comprehensive set of structured information. Thus processing conjunctive sentences is important to improve the performance of Open IE models.

It is a common practice to apply constituency parsers or dependency parsers to identify the coordination structures of a sentence. However, there are several drawbacks. First, the state-of-the-art syntactic parsers confront an increase of errors when processing conjunctive sentences, especially when the input sentence contains complex coordination structures. Second, applying parsers can be slow, which will make the identification of coordination less efficient. Existing coordination boundary detection methods rely on the results of syntactic parsers (Ficler and Goldberg, 2016, 2017; Saha and Mausam, 2018) and thus still face similar drawbacks.

In this work, we approach the coordination recognition problem without using syntactic parsers and propose a simple yet effective pipeline model COordination RECognizer (CoRec). CoRec composes of two steps: coordinator identification and conjunct boundary detection. For coordinator identification, we consider three types of coordinator spans: contiguous span coordinators (e.g. 'or' and 'as well as'), paired span coordinators (e.g. 'either...or...'), and coordination with 'respectively'. Given each identified coordinator span, we formulate the conjunct boundary detection task as a sequence labeling task and design a position-aware BIOC labeling schema based on the unique characteristics of this task. We also present a simple trick called coordinator markers that can greatly improve the model performance.

Despite CoRec's simplicity, we find it to be both effective and efficient in the empirical studies: CoRec consistently outperforms state-of-the-art models on benchmark datasets from both general domain and biomedical domain. Further experiments demonstrate that processing the conjunctive sentences with CoRec can enhance the yield of Open IE models.

In summary, our main contributions are:

- We propose a pipeline model CoRec, a special-

---

[1] https://github.com/qingwang-isu/CoRec

ized coordination recognizer without using syntactic parsers.

- We formulate the conjunct boundary detection task as a sequence labeling task with position-aware labeling schema.
- Empirical studies on three benchmark datasets from various domains demonstrate the efficiency and effectiveness of CoRec, and its impact on yield of Open IE models.

## 2 Related Work

For the tasks of coordination boundary detection and disambiguation, earlier heuristic, non-learning-based approaches design different types of features and principles based on syntactic and lexical analysis (Hogan, 2007; Shimbo and Hara, 2007; Hara et al., 2009; Hanamoto et al., 2012; Corro and Gemulla, 2013). Ficler and Goldberg (2016) are the first to propose a neural-network-based model for coordination boundary detection. This model operates on top of the constituency parse trees, and decomposes the trees to capture the syntactic context of each word. Teranishi et al. (2017, 2019) design similarity and replaceability feature vectors and train scoring models to evaluate the possible boundary pairs of the conjuncts. Since these methods are designed to work on conjunct pairs, they have natural shortcomings to handle more than two conjuncts in one coordination.

Researchers in the Open Information Extraction domain also consider coordination analysis to be important to improve model performance. CALM, proposed by Saha and Mausam (2018), improves upon the conjuncts identified from dependency parsers. It ranks conjunct spans based on the 'replaceability' principle and uses various linguistic constraints to additionally restrict the search space. OpenIE6 (Kolluru et al., 2020) also has a coordination analyzer called IGL-CA, which utilizes a novel iterative labeling-based architecture. However, its labels only focus on the boundaries of the whole coordination and do not utilize the position information of the specific conjuncts.

## 3 Methodology

### 3.1 Task Formulation

Given a sentence $S = \{x_1, ..., x_n\}$, we decompose the coordination recognition task into two sub-tasks, coordinator identification and conjunct boundary detection. The coordinator identifier aims to detect all potential target coordinator spans

from $S$. The conjunct boundary detector takes the positions of all the potential target coordinator spans as additional input and detects the conjuncts coordinated by each target coordinator span.

### 3.2 Label Formulation

Since the coordinator spans are usually short, we adopt simple binary labels for the coordinator identification sub-task: label 'C' for tokens inside coordinator spans and 'O' for all other tokens.

For the conjunct boundary detection sub-task, conjuncts can be long and more complicated. Thus we formulate this sub-task as a sequence labeling task. Specifically, inspired by the BIO (Beginning-Inside-Outside) (Ramshaw and Marcus, 1995) labeling schema of the NER task, we also design a position-aware labeling schema, as previous researches have shown that using a more expressive labeling schema can improve model performance (Ratinov and Roth, 2009; Dai et al., 2015).

The proposed labeling schema contains both position information for each conjunct and position information for each coordination. For each conjunct, we use 'B' to label the beginning token and 'I' to label the following tokens. For each coordination structure, we further append 'before' and 'after' tags to indicate the relative positions to the target coordinator, which is/are labeled as 'C'.

### 3.3 Labeling Details

In this section, we categorize the coordinator spans into three types and use simple examples to show the labeling details of each type.

**Contiguous Span Coordinators** Processing contiguous span coordinator is straightforward. Take the sentence "*My sister likes apples, pears, and grapes.*" as an example, following Section 3.2 we should generate one instance with labels as shown in Figure 1 (1).

**Paired Span Coordinators** Each paired span coordinator consists of two coordinator spans: the left coordinator span, which appears at the beginning of the coordination, and the right coordinator span, which appears in the middle. The right coordinator span stays more connected with the conjuncts due to relative position. Therefore, we choose to detect the conjuncts only when targeting the right coordinator span. Take the sentence "*She can have either green tea or hot chocolate.*" as an example, following Section 3.2 we should generate two instances with labels as shown in Figure 1 (2).

**Figure 1 — Labeling Examples**

(1):

| My | sister | likes | apples | , | pears | , | [C] | and | [C] | grapes | . |
|---|---|---|---|---|---|---|---|---|---|---|---|
| O | O | O | B-before | I-before | B-before | I-before | C | C | C | B-after | O |

(2):

| She | can | have | [C] | either | [C] | green | tea | or | hot | chocolate | . |
|---|---|---|---|---|---|---|---|---|---|---|---|
| O | O | O | C | C | C | O | O | O | O | O | O |
| She | can | have | either | green | tea | [C] | or | [C] | hot | chocolate | . |
| O | O | O | C | B-before | I-before | C | C | C | B-after | I-after | O |

(3):

| The | dog | [C] | and | [C] | the | cat | were | named | Jack | and | Sam | respectively | . |
|---|---|---|---|---|---|---|---|---|---|---|---|---|---|
| B-before | I-before | C | C | C | B-after | I-after | O | O | O | O | O | O | O |
| The | dog | and | the | cat | were | named | Jack | [C] | and | [C] | Sam | respectively | . |
| O | O | O | O | O | O | O | B-before | C | C | C | B-after | O | O |
| The | dog | and | the | cat | were | named | Jack | and | Sam | [C] | respectively | [C] | . |
| B-before | I-before | C | B-after | I-after | O | O | B-before | C | B-after | C | C | C | O |

(4):

| That | should | stop | a | lot | of | account | - | churning | [C] | and | [C] | produce | a |
|---|---|---|---|---|---|---|---|---|---|---|---|---|---|
| O | O | B-before | I-before | I-before | I-before | I-before | I-before | I-before | C | C | C | B-after | I-after |
| stock | market | driven | only | by | professional | concern | , | careful | thought | and | good | sense | . |
| I-after | I-after | I-after | I-after | I-after | I-after | I-after | I-after | I-after | I-after | I-after | I-after | I-after | O |
| That | should | stop | a | lot | of | account | - | churning | and | produce | a | stock | market |
| O | O | O | O | O | O | O | O | O | O | O | O | O | O |
| driven | only | by | professional | concern | , | careful | thought | [C] | and | [C] | good | sense | . |
| O | O | O | B-before | I-before | I-before | B-before | I-before | C | C | C | B-after | I-after | O |

Figure 1: Labeling Examples. (1): Conjunctive sentence labeling with contiguous span coordinator 'and'; (2): Conjunctive sentence labeling with paired span coordinator 'either...or...'; (3): Conjunctive sentence labeling with 'respectively'; (4): Nested conjunctive sentence labeling.

**Coordination with 'Respectively'** The conjunctive sentence with 'respectively' usually has the structure '...and...and...respectively...', where the first and the second coordination have the same number of conjuncts.

The 'respectively' case is handled differently in training and inference. During training, for a '...and...and...respectively...' sentence, we recognize three coordinator spans ('and', 'and', and 're-spectively') and generate three training instances with different target coordinator spans. Take the sentence "*The dog and the cat were named Jack and Sam respectively.*" as an example, we should generate three instances with labels as shown in Figure 1 (3). This is because 'and' is one of the most common coordinators and occurs much more frequently than '...and...and...respectively...'. If we only consider "respectively" as the sole target co-ordinator in the sentence and do not consider 'and' as a coordinator during training, the model can be confused. During inference, when encountering a sentence containing 'respectively', we consider the conjuncts recognized when 'respectively' is the target coordinator span as the final result.

### 3.4 Coordinator Identifier

As mentioned above, the coordinator identification sub-task is formulated as a binary classification problem. Our coordinator identifier uses a BERT (Devlin et al., 2019) encoder to encode a sentence $S = \{x_1, x_2, ..., x_n\}$, and the output is:

$$[\boldsymbol{h}_1^c, ..., \boldsymbol{h}_n^c] = Enc_1([x_1, ..., x_n]). \quad (1)$$

A linear projection layer is then added. We denote coordinator spans detected by the coordinator identifier as $P = \{p_1, p_2, ..., p_k\}$.

### 3.5 Conjunct Boundary Detector

The conjunct boundary detector then processes each target coordinator span $p_t \in P$ independently to find all coordinated conjuncts in sentence $S$.

To inject the target coordinator span information into the encoder, we insert coordinator markers, '[C]' token, before and after the target coordinator span, respectively. The resulting sequence is $S_m = \{x_1, ..., [C], p_t, [C]..., x_n\}$. For simplicity we denote $S_m = \{w_1, ..., w_m\}$.

The marked sequence $S_m$ is fed into a BERT encoder:

$$[\boldsymbol{h}_1^{cbd}, ..., \boldsymbol{h}_m^{cbd}] = Enc_2([w_1, ..., w_m]). \quad (2)$$

The position information of all the coordinators found by the coordinator identifier can help the model to understand the sentence structure. Thus we encode such information into a vector $\boldsymbol{b_i}$ to indicate if $w_i$ is part of a detected coordinator span. Given $w_i \in S_m$, we concatenate its encoder output and coordinator position encoding as $\boldsymbol{h}_i^o = [\boldsymbol{h}_i^{cbd}; \boldsymbol{b}_i]$.

Finally, we use a CRF (Lafferty et al., 2001) layer to ensure the constraints on the sequential rules of labels and decode the best path in all possible label paths.

### 3.6 Training & Inference

The coordinator identifier and the conjunct boundary detector are trained using task-specific losses.

For both, we fine-tune the two pre-trained $BERT_{base}$ encoders. Specifically, we use cross-entropy loss:

$$\mathcal{L}_c = -\sum_{x_i \in S} logP_c(t_i^*|x_i), \qquad (3)$$

$$\mathcal{L}_{cbd} = -\sum_{w_i \in S_m} logP_{cbd}(z_i^*|w_i), \qquad (4)$$

where $t_i^*$, $z_i^*$ represent the ground truth labels. During inference, we first apply the coordinator identifier and obtain:

$$y_c(x_i) = \underset{t_i \in T}{\text{argmax}}\, P_c(t_i|x_i). \qquad (5)$$

Then we use its prediction $y_c(x_i)$ with the original sentence as input to the conjunct boundary detector and obtain:

$$\boldsymbol{y} = \underset{[z_1,...,z_m],z_i \in Z}{\text{argmax}}\, P_{cbd}([z_1, ..., z_m]|[w_1, ..., w_m]), \qquad (6)$$

where $T$ and $Z$ represent the set of possible labels of each model respectively.

### 3.7 Data Augmentation

We further automatically augment the training data. The new sentences are generated following the symmetry rule, by switching the first and last conjuncts of each original training sentence. Since all sentences are augmented once, the new data distribution only slightly differs from the original one, which will not lead to a deterioration in performance (Xie et al., 2020).

## 4 Experiments

**Training Setup**  The proposed CoRec is trained on the training set (WSJ 0-18) of Penn Treebank[2] (Marcus et al., 1993) following the most common split, and WSJ 19-21 are used for validation and WSJ 22-24 for testing. The ground truth constituency parse trees containing coordination structures are pre-processed to generate labels for the two sub-tasks as follows. If a constituent is tagged with 'CC' or 'CONJP', then it is considered a co-ordinator span. For each coordinator span, we first extract the constituents which are siblings to the coordinator span, and each constituent is regarded as a conjunct coordinated by that coordinator span. We automatically generate labels as described in Section 3.2. We also manually check and correct labels for complicated cases.

**Testing Setup**  We use three testing datasets to evaluate the performance of the proposed CoRec model. The first dataset, ontoNotes, contains 1,000 randomly selected conjunctive sentences from the English portion of OntoNotes Release 5.0[3] (Weischedel et al., 2013). The second dataset, Genia, contains 802 conjunctive sentences from the testing set of GENIA[4] (Ohta et al., 2002), a benchmark dataset from biomedical domain. The third dataset, Penn, contains 1,625 conjunctive sentences from Penn Treebank testing set (WSJ 22-24). These three datasets contain the gold standard constituency parsing annotations. We convert them into the OC and BIOC labels in the same way as described in Section 4.

Each testing dataset is further split into a 'Simple' set and a 'Complex' set based on the complexity of the coordinators. 'Simple' set contains instances with 'and', 'but', 'or' as target coordinators only and these three coordinators can be handled by all baselines. Whereas the 'Complex' set contains other scenarios including multi-token contiguous span coordinators (e.g. 'as well as'), the paired span coordinators (e.g., 'not only...but also...'), and coordination with 'respectively'. 'Complex' instances may not be hard to predict. For example, the instances with paired span coordinators may be easier for the model since the first coordinator span may give better clues about the boundaries of conjuncts. Table 1 provides the respective counts of instances in 'Simple' and 'Complex' sets for three testing datasets.

|         | ontoNotes | Genia | Penn |
|---------|-----------|-------|------|
| Simple  | 1123      | 2193  | 1981 |
| Complex | 127       | 327   | 267  |

Table 1: The statistics of 'Simple' and 'Complex' sets on three testing datasets.

**Baseline Methods**  We compare the proposed CoRec with two categories of baseline methods: parsing-based and learning-based methods. Parsing-based methods convert the constituency parsing results and regard constituents at the same depth with the target coordinator spans as coordinated conjuncts. We adopt two state-of-the-art constituency parsers, AllenNLP (Joshi et al., 2018) and

---

[2]https://catalog.ldc.upenn.edu/LDC99T42

[3]https://catalog.ldc.upenn.edu/LDC2013T19
[4]http://www.geniaproject.org/genia-corpus/treebank

| Model | ontoNotes (Simple) | | | | Genia (Simple) | | | | Penn (Simple) | | | |
|---|---|---|---|---|---|---|---|---|---|---|---|---|
| | **P** | **R** | **F1** | **Time** | **P** | **R** | **F1** | **Time** | **P** | **R** | **F1** | **Time** |
| AllenNLP | 74.2 | 68.4 | 71.2 | 452s | 79.7 | 47.7 | 59.7 | 1059s | **88.7** | 67.7 | 76.8 | 823s |
| Stanford | 56.9 | 53.4 | 55.1 | 763s | 73.8 | 72.2 | 73.0 | 1722s | 81.8 | 79.3 | 80.6 | 1387s |
| Teranishi+19 | 68.3 | 60.8 | 64.7 | 167s | 76.4 | 65.2 | 70.3 | 136 s | 74.2 | 75.5 | 75.4 | 217s |
| IGL-CA | **77.6** | 59.7 | 67.5 | **17s** | 78.0 | 64.3 | 71.0 | 27s | 87.9 | 86.9 | 87.4 | **17s** |
| CoRec (our) | 72.4 | **75.8** | **74.1** | 32s | **82.0** | **81.2** | **81.6** | **15s** | 88.3 | **89.2** | **88.8** | 57s |
| | ontoNotes (Complex) | | | | Genia (Complex) | | | | Penn (Complex) | | | |
| AllenNLP | **84.6** | 49.4 | 62.4 | 105s | **82.3** | 25.7 | 39.2 | 370s | 90.2 | 61.7 | 73.2 | 363 s |
| Stanford | 62.4 | 34.5 | 44.4 | 248 s | 64.3 | 32.9 | 43.5 | 831s | 86.1 | 69.7 | 77.1 | 530 s |
| CoRec (our) | 73.1 | **79.3** | **76.0** | **4s** | 67.7 | **56.7** | **61.7** | **9s** | **91.5** | **89.5** | **90.5** | **10s** |

Table 2: Performance Comparison (average over 5 runs)

Stanford CoreNLP [5] parsers, for this category. For learning-based methods, we choose two state-of-the-art models for coordination boundary detection, Teranishi+19 (Teranishi et al., 2019), and IGL-CA (Kolluru et al., 2020). All results are obtained using their official released code.

**Evaluation Metrics**   We evaluate both the effectiveness and efficiency of different methods. We evaluate effectiveness using span level precision, recall, and F1 scores. A predicted span is correct only if it is an exact match of the corresponding span in the ground truth.

For efficiency evaluation, we report the inference time of each method. All experiments are conducted on a computer with Intel(R) Core(TM) i7-11700k 3.60GHz CPU, NVIDIA(R) RTX(TM) 3070 GPU, and 40GB memory.

## 4.1   Main Results

The results are shown in Table 2. In terms of effectiveness, CoRec's recall and F1 scores are higher than all baselines on all datasets, and the improvement on F1 scores is 2.9, 8.6, and 1.4 for ontoNotes, Genia, and Penn compared to the best baseline methods, respectively. Although CoRec is not trained on a biomedical corpus, it still demonstrates superior performance. The inference time of CoRec is also competitive.

## 4.2   Impact of CoRec on Open IE Tasks

To show the impact of CoRec on downstream tasks, we implement a sentence splitter that generates simple, non-conjunctive sub-sentences from CoRec's output. We apply two state-of-the-art Open IE models, Stanford OpenIE (Angeli et al., 2015) and OpenIE6 (Kolluru et al., 2020), to extract unique relation triples on the Penn dataset before and after our sentence splitting. The results are shown in Table 3.

[5] https://nlp.stanford.edu/software/srparser.html

| Model | Before | After | Yield |
|---|---|---|---|
| Stanford | 12210 | 21284 | +74.3% |
| OpenIE6 | 8084 | 12085 | +58.4% |

Table 3: The impact of CoRec on Open IE Yield

| Model | Precision | Recall | F1 |
|---|---|---|---|
| BERT | 78.73 | 85.46 | 81.96 |
| +[C] mark | 87.15 | 88.92 | 88.03 |
| +CRF | 88.36 | 89.35 | 88.85 |
| +aug | **89.28** | **90.21** | **89.74** |

Table 4: Ablation Study

Sentence splitting significantly increases the yield of unique extractions for both models. Though OpenIE6 implements the coordination boundary detection method IGL-CA, the coordination structure still negatively impacts the Open IE yield.

## 4.3   Ablation Study

We perform an ablation study to assess the contribution of different components to performance. The base model only uses BERT encoder, then coordinator markers, CRF, and data augmentation are incrementally added. The testing results on Penn dataset are shown in Table 4. From the results, we can see that all of the components can improve the performance in terms of precision and recall.

## 5   Conclusions

In this paper, we develop CoRec, a simple yet effective coordination recognizer without using syntactic parsers. We approach the problem by formulating a pipeline of coordinator identification and conjunct boundary detection. CoRec can not only detect the boundaries of more than two coordinated conjuncts but also handle multiple coordinations in one sentence. It can also deal with both simple and complex cases of coordination. Experiments show that CoRec outperforms state-of-the-art models on datasets from various domains. Further experiments imply that CoRec can improve the yield of state-of-the-art Open IE models.

## Limitations

**Language Limitation** The proposed CoRec model works mostly for languages with limited morphology, like English. Our conclusions may not be generalized to all languages.

**Label Quality Limitation** We use ground truth constituency parse trees from Penn Treebank, GE-NIA, and OntoNotes Release 5.0 (Marcus et al., 1993; Weischedel et al., 2013; Ohta et al., 2002) to generate the labels. Since the parsing does not target for the coordination recognition task, we apply rules for the conversion. A single author inspected the labels for complicated cases but did not inspect all the labels. There could be erroneous labels in the training and testing data.

**Comparison Limitation** Comparison to the parsing-based methods may not be precise. Parsers are not specialized in the coordination recognition task. Our task and datasets may not be the best fit for their models.

## Ethics Statement

We comply with the EMNLP Code of Ethics.

## Acknowledgments

The work was supported in part by the US National Science Foundation under grant NSF-CAREER 2237831. We also want to thank the anonymous reviewers for their helpful comments.

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

## A  Error Analysis

To better understand the bottleneck of CoRec, we conduct a case study to investigate the errors that CoRec makes. We randomly selected 50 wrong predictions and analyzed their reasons. We identify four major types of errors as follows (for detailed examples check Table 5):

**Ambiguous Boundaries (38%)**  The majority of the errors occurred when the detected boundaries are ambiguous. In this case, although our prediction is different from the gold standard result, they can both be treated as true. We identify two common types of ambiguous boundaries: ambiguous shared heads (28%) and ambiguous shared tails (10%). The former is usually signaled by 'a/an/the' and shared modifiers. The latter is usually signaled by prepositional phrases.

**Errors without Obvious Reasons (32%)**  Many errors occurred without obvious reasons. However, we observe that CoRec makes more mistakes when the original sentences contain a large amount of certain symbols (e.g., '-', '.').

**Wrong Number of Conjuncts (22%)**  Sometimes CoRec detects most conjuncts in the gold standard set but misses a few conjuncts. In some other cases, CoRec would detect additional conjuncts to the correct conjuncts.

**Low-Quality Gold Labels (8%)**  We find there may also be some low-quality ground truth parse trees, thus generating incorrect gold labels. In this case CoRec may make a correct prediction that is different from the ground truth.

| Category | Ground Truth | CoRec |
|---|---|---|
| Ambiguous Boundaries | I'm not going to worry about one day's decline, said Kenneth Olsen, digital equipment corp. president, who was leisurely strolling through the bright [orange] and [yellow] leaves of the mountains here after his company's shares plunged $5.75 to close at $86.5. | I'm not going to worry about one day's decline, said Kenneth Olsen, digital equipment corp. president, who was leisurely strolling through [the bright orange] and [yellow] leaves of the mountains here after his company's shares plunged $5.75 to close at $86.5. |
| Errors without Obvious Reasons | For example, the delay in selling people's heritage savings, Salina Kan, with $1.7 billion in assets, has forced the RTC to consider selling off the thrift [branch-by-branch,] instead of [as a whole institution]. | For example, the delay in selling people's heritage savings, Salina Kan, with $1.7 billion in assets, has forced the RTC to consider [selling off the thrift branch-by-branch,] instead of [as a whole institution]. |
| Wrong Number of Conjuncts | Sales of Pfizer's important drugs, [Feldene for treating arthritis,] and [Procardia, a heart medicine], have shrunk because of increased competition. | Sales of [Pfizer's important drugs,] [Feldene for treating arthritis,] and [Procardia, a heart medicine], have shrunk because of increased competition. |
| Low-Quality Gold Labels | The executives were remarkably unperturbed by the plunge even though it [lopped billions of dollars off the value of their companies] and [millions off their personal fortunes]. | The executives were remarkably unperturbed by the plunge even though it lopped [billions of dollars off the value of their companies] and [millions off their personal fortunes]. |

Table 5: Case study of conjunct boundary detection results on the Penn dataset. For each case, the ground truth conjuncts are colored red and the CoRec detected conjuncts are colored blue.