# OpenReview forum: "CoRec: An Easy Approach for Coordination Recognition"
_EMNLP/2023/Conference — EMNLP 2023 Main_

### Official Review · Reviewer_12p3 · 2023-08-03

**Soundness:** 4

**Excitement:**

3: Ambivalent: It has merits (e.g., it reports state-of-the-art results, the idea is nice), but there are key weaknesses (e.g., it describes incremental work), and it can significantly benefit from another round of revision. However, I won't object to accepting it if my co-reviewers champion it.

**Paper Topic And Main Contributions:**

This paper presents a method for coordination structure analysis based on BERT.
By formulating the conjunct boundary detection as a sequence labeling problem, unlike previous methods, it handles a coordination including more than two conjuncts straightforwardly and efficiently.
It brought modest improvement on OntoNote and PennWSJ data, and significant improvement on Genia Corpus.

**Reasons To Accept:**

The accuracy improvement on the Genia data is significant.

**Reasons To Reject:**

Description of the method is not satisfactorily thorough even considering the space limitation.
It is not imaginable how to handle the conjuncts of the paired span coordinators and coordination with “respectively" without referring to the appendix. The case for “respectively" is not clear to me even after reading the appendix. How the three sequence labeling results in Table 7 are merged (if the predictions are not consistent)?

**Reproducibility:**

3: Could reproduce the results with some difficulty. The settings of parameters are underspecified or subjectively determined; the training/evaluation data are not widely available.

**Reviewer Confidence:**

4: Quite sure. I tried to check the important points carefully. It's unlikely, though conceivable, that I missed something that should affect my ratings.

**Typos Grammar Style And Presentation Improvements:**

* L214: the the -> the
* L252: in 4 -> in Section 4

---

> ### Author Rebuttal · Authors · 2023-08-27
>
> Thank you for your reviews and questions.
> The “respectively” case is handled differently in training and inference.
> During inference, when encountering a sentence containing “respectively”, we consider the conjuncts recognized when “respectively” is the target coordinator span as the final result.
> During training, for “and … and … respectively” sentence, we recognize three coordinator spans (“and”, “and”, and “respectively”) and generate three training instances with different target coordinator spans (as shown in Table 7). This is because “and” is a one the most common coordinators and occurs much more frequently than “and … and … respectively”. If we only consider “respectively” as the sole target coordinator in the sentence and do not consider “and” as a coordinator during training, the model can be confused.
> We apologize for any confusion caused and we will enhance the clarity of relevant descriptions in the final version of the paper.

---

### Official Review · Reviewer_r3Vf · 2023-08-03

**Soundness:** 5

**Excitement:**

4: Strong: This paper deepens the understanding of some phenomenon or lowers the barriers to an existing research direction.

**Paper Topic And Main Contributions:**

This paper presents a process for the identification and demarcation of conjunctive phrases in open text. Having accurate conjunctions is important for open information extraction to establish scope and clear bounds for what the IE system is targeting. Their system, CoRec, is a cascade of two independently trained processes. The training is based on the Penn Treebank with the usual split. The first process identifies the conjuncts ('coordinators') and introduces markers around their span. The second process employs BIO labels on constituents and position information derived from the training. Their model is built using a fine-tuned BERT encoder.

In addition to simple conjuncts such as 'and', 'but' or 'as well as'.  Their system handles paired conjunctions: 'either .. or', and the 'respectively' construction. In addition to Treebank, they test their system on OntoNotes and the GENIA biology data set, consistently outperforming their parsing-based baselines and selected earlier learning based methods.

**Questions For The Authors:**

A. Have you considered conjunction over larger constituents such as verb phrases or clauses ("J likes games and K does too") ?

B. Could you elaborate on what goes into the runtimes you report? You assert in your introduction that parsers are slow, which is one of the reasons you took the direction you did. Just what is being measured in the times reported in Table 1. "Inference time" is not a transparent concept. What unit of text is it being measured over and what are you counting? OpenIE6, for example, reports speed as sentences per second (its table 2).
Re. author's response. I was looking a timing definition that was more like OpenIE6's: some quantity of text tokens per unit of time. Saying that it reflects the time to get through a particular data set just pushes the question further back to what is in the data set.

**Reasons To Accept:**

-- Makes judicious use of just enough LLM technology to process its training data and uses a clear cascading architecture of straightforward components rather than
 trying a complex custom transformer architecture.

-- Quite clearly written, with a sensible split between the body argument and details in an short appendix (all in 8 pages).

-- Includes an informative error analysis and a persuasive ablation study.

**Reasons To Reject:**

I see no reason to reject this paper

**Reproducibility:**

5: Could easily reproduce the results.

**Reviewer Confidence:**

3: Pretty sure, but there's a chance I missed something. Although I have a good feel for this area in general, I did not carefully check the paper's details, e.g., the math, experimental design, or novelty.

**Typos Grammar Style And Presentation Improvements:**

Overall this is paper is clearly written. This reviewer's only complaint is the notation is the technical core of the paper (sect. 3.3 through 3.5) should be introduced a bit more slowly, noting the meaning of the symbols rather than requiring us to puzzle them out, e.g. the 'cbd' superscript.

---

> ### Author Rebuttal · Authors · 2023-08-27
>
> We really appreciate your reviews and suggestions.
> 1. (A.) We have considered conjunctions over larger constituents such as verb phrases and clauses.
> A pertinent example can be found in the Genia testing dataset: “Our results suggest [that CaMKIV may represent a physiologically relevant CREB kinase in T cells] and [that the enzyme is also required to ensure the normal expansion of T cells in the thymus].”
> We have also considered nested coordination such as this example “That should [stop a lot of account-churning] and [produce a stock market driven only by {professional concern}, {careful thought} and {good sense}].” from Penn Treebank WSJ (we use [] and {} to highlight the conjunct boundaries).
> The average lengths of conjuncts in ontoNotes, Genia, and Penn are 4.87, 4.78, and 5.60, respectively.
> 2. (B.) The reported runtimes correspond to the number of seconds required for each specific testing dataset to be completed (get the conjuncts detected) by each method.

---

### Official Review · Reviewer_4paW · 2023-08-07

**Typos Grammar Style And Presentation Improvements:** 1. L240/Table 1, ontoNotes -> OntoNotes
**Soundness:** 4

**Excitement:**

3: Ambivalent: It has merits (e.g., it reports state-of-the-art results, the idea is nice), but there are key weaknesses (e.g., it describes incremental work), and it can significantly benefit from another round of revision. However, I won't object to accepting it if my co-reviewers champion it.

**Paper Topic And Main Contributions:**

Coordination recognition refers to a task that identifies the coordination structure in sentences, which is beneficial for a variety of NLP tasks, including Open IE. In contrast to previous research that relies on parser outputs, this paper proposes a two-step solution. The first step is to identify coordinators and the second step solves the structure by finding relevant arguments from its neighbors. Experiments show that the CRF model helps find the boundaries of coordination structures, present that the model achieves the sota scores across three benchmark datasets, and demonstrates the efficiency of the model. Additionally, the split sentences that generate from model predictions contribute to the downstream NLP task, Open IE.

**Questions For The Authors:**

1. I can understand why the ablation study section is too short. I'm wondering if the author can present the scores just using CRF, unifying the two-step architecture into one CRF, and show how much degradation is in the ablation study? This can also answer why the two-step model performs better than other learning-based models.
2. Among "errors without obvious reasons", how many are associated with ellipsis (like the example in Table 8)? Can this error be counted as one major error type?

**Reasons To Accept:**

1. The paper avoids the existing parser's problems with coordination and provides a better solution to recognize coordination structures. The model achieves the sota performance on three benchmarks.
2. The model exhibits superior efficiency when compared to other models.

**Reasons To Reject:**

1. The task definition is not straightforward and clear to audiences. What is the extent of coordination for this task —— does it refer to intra-sentential coordination that connects nominal phrases or verbal phrases, or does it also include inter-sentential discourse relations? Can authors also provide an example on page 1 or 2 to illustrate the coordination recognition task?
2. The classification of "simple" and "complex" coordinators (or maybe just their names) is not clear to me. If "complex" coordinators may not be hard to predict, I would be confused with its name when reading the paper.
3. Please refer to questions.

**Reproducibility:**

4: Could mostly reproduce the results, but there may be some variation because of sample variance or minor variations in their interpretation of the protocol or method.

**Reviewer Confidence:**

3: Pretty sure, but there's a chance I missed something. Although I have a good feel for this area in general, I did not carefully check the paper's details, e.g., the math, experimental design, or novelty.

---

> ### Author Rebuttal · Authors · 2023-08-27
>
> Thank you for taking the time to review our work.
>
> Response to Reasons to Reject:
> 1. We consider coordinating conjunctions that join two or more elements of equal grammatical rank and syntactic importance, which can be noun phrases, verb phrases, clauses, etc.
> Thank you for your suggestion and we will incorporate an example to illustrate the coordination recognition task in the final version of the paper.
> 2. Both the “Simple” and “Complex” refer to the complexity of the coordinators, rather than indicating levels of difficulty. “Simple” set contains “and”, “but”, “or” as target coordinators only. These three coordinators can be handled by all baselines. Whereas “Complex” set contains other coordinators, including multi-token coordinators (e.g. “as well as”) and disjoint coordinators (e.g. “not only...but also...”). For a more comprehensive list of “complex” cases, kindly refer to Appendix B. Learning-based coordination recognition baseline methods either do not recognize these cases as coordinators (e.g. “as well as”), or treat only a single token as the coordinator (e.g. only “but” in “not only...but also...” is considered as the coordinator).
>
> Response to Questions:
> 1. Does the “two-step architecture” refer to recognizing coordinators and recognizing conjuncts in two separate steps?
> The reason we do not unify these two steps into one CRF is that we also need to handle sentences with nested coordination such as this example “That should [stop a lot of account-churning] and [produce a stock market driven only by {professional concern}, {careful thought} and {good sense}].” from Penn Treebank WSJ (we use [] and {} to highlight the conjunct boundaries). To handle the nested coordination, the target coordinator needs to be specified first.
> 2. Out of the 16 sentences categorized as "errors without obvious reasons", only 2 sentences are associated with ellipsis. Given this occurrence, these error instances might not be considered as a major error type.
> The following is another example from this category:
> Sentence: Just the fact 0 we ' re on 24 hours is no longer bulletin , ' ' says Ed Turner , CNN ' s executive vice president , news gathering - lrb - and no relation to Ted Turner - rrb - .
> Prediction: Just the fact 0 we ' re on 24 hours is no longer bulletin , ' ' says Ed Turner , CNN ' s executive vice president , news gathering - lrb [-] and [no relation to Ted Turner -] rrb - .
> We use [] to highlight the predicted conjunct boundaries.
> We suspect the special symbols (- lrb - and - rrb - refer to left and right round brackets, respectively) in this sentence might cause misunderstandings of BERT.
> Overall, we did not find obvious patterns for those errors. Therefore, we categorized them as "errors without obvious reasons".

---

### Meta-Review · Area_Chair_N2No · 2023-09-15

**Recommendation:** 4

**Metareview:**

This paper proposes a new approach to conjunction recognition. The method consists of two steps: (1) identification of coordinators and (2) filling in relevant arguments through sequence labeling. The method outperforms regular parsers on three datasets and improves the performance of the downstream task (information extraction).

The reviewers agreed while ranking **soundness** of this paper as strong (scores 4, 4, and 5). Among the strengths of this submission, they listed a *straightforward architecture* and *informative error analysis*.
In the first round, the reviewers had many questions regarding the details of the proposed method. However, the answers from the authors clarified most of the issues. Therefore, it is strongly recommended that these explanations and clarifications be included in the camera-ready version.

Regarding **excitement**, the reviewers chose more neutral scores (3, 3, and 4). On the one hand, they were convinced that the paper provides a *good solution* to the described problem. The proposed model achieves *state-of-the-art performance* and tackles concrete shortcomings of other approaches. On the other hand, conjunction recognition is a very specialized task. Therefore, the paper needs to be clear and provide all the necessary details to reach a broad audience. The reviewers found that this level of detail needs to be improved.

To sum up, it is a sound submission about a particular task. It requires clarification and details that can be added in the camera-ready version. When the paper is clearer, it will also make its readers more excited.

---

### Decision · Program_Chairs · 2023-10-07

**Decision:**

Accept-Main

**Comment:**

This paper proposes a new approach to conjunction recognition. The method consists of two steps: (1) identification of coordinators and (2) filling in relevant arguments through sequence labeling. The method outperforms regular parsers on three datasets and improves the performance of the downstream task (information extraction).

The reviewers agreed while ranking **soundness** of this paper as strong (scores 4, 4, and 5). Among the strengths of this submission, they listed a *straightforward architecture* and *informative error analysis*.
In the first round, the reviewers had many questions regarding the details of the proposed method. However, the answers from the authors clarified most of the issues. Therefore, it is strongly recommended that these explanations and clarifications be included in the camera-ready version.

Regarding **excitement**, the reviewers chose more neutral scores (3, 3, and 4). On the one hand, they were convinced that the paper provides a *good solution* to the described problem. The proposed model achieves *state-of-the-art performance* and tackles concrete shortcomings of other approaches. On the other hand, conjunction recognition is a very specialized task. Therefore, the paper needs to be clear and provide all the necessary details to reach a broad audience. The reviewers found that this level of detail needs to be improved.

To sum up, it is a sound submission about a particular task. It requires clarification and details that can be added in the camera-ready version. When the paper is clearer, it will also make its readers more excited.